# Photo-induced stress relaxation in reconfigurable disulfide-crosslinked supramolecular films visualized by dynamic wrinkling

Shuzhen Yan [1,3], Kaiming Hu [2,3], Shuai Chen [1], Tiantian Li [1], Wenming Zhang[2] ✉, Jie Yin[1] & Xuesong Jiang [1] ✉

Stress relaxation in reconfigurable supramolecular polymer networks is strongly related to intermolecular behavior. However, the relationship between molecular motion and macroscopic mechanics is usually vague, and the visualization of internal stress reflecting precise regulation of molecules remains challenging. Here, we present a strategy for visualizing photo-driven stress relaxation induced by infinitesimal perturbations in the intermolecular exchange reaction via reprogrammable wrinkle patterns. The supramolecular films exhibit visible changes in microscopic wrinkle topography through ultraviolet (UV)-induced dynamic disulfide exchange reaction. In accordance with the trans-scale theoretical models, which quantitatively evaluate the chemical-dependent mechanical stresses in the supramolecular network, the unexposed disordered wrinkles evolved into highly oriented patterns and underwent subsequent mutations after thermal treatment. The stress-sensitive wrinkle macro-patterns can be repetitively written/erased through network topology rearrangement using different stimuli. This strategy provides an approach for visualizing and understanding the molecular behavior from dynamic chemistry to mechanical changes, and directly programming wrinkle patterns with regulated structures.

In chaos theory, the "butterfly effect" is the sensitive dependence on initial conditions, whereby significant results are influenced by the most insignificant perturbations[1–4]. Similar to such interesting phenomenon, the polymer chain reactions triggered by original molecular changes have the potential to affect the functional properties and applications of polymeric materials[3,5,6]. Even a slight chain scission or formation leads to rather notable performance transformation; for instance, the hydrogen bonds endow the crosslinked hydrogel with ultra-strong mechanical properties[7–9], and imine bonds can undergo thermal transamination to realize thermoset reprocessing of thermoplastics[10,11]. Hence, regulation of polymer performance on demand using appropriate molecular chemistry opens multiple pathways for realizing tunable and responsive smart materials. At the same time, establishing a chemical-mechanical relationship from the intermolecular behavior on a molecular scale to macroscale properties of polymeric networks has been extremely challenging to date. Moreover, facile and robust visual methods for reflecting the overlooked intermolecular perturbations are rarely reported, even though they are highly needed.

Recently, sensitive patterned surfaces have attracted increasing attention because of their dynamic micro-/nano-structures which are promising for applications in smart displays[12–15], structural colors[16–19], electronic devices[20–22], microfluidic channels[23,24], and interface engineering[22,25,26]. Lin's group first demonstrated polarized light-

[1]School of Chemistry & Chemical Engineering, Frontiers Science Center for Transformative Molecules, State Key Laboratory for Metal Matrix Composite Materials, Shanghai Jiao Tong University, Shanghai 200240, PR China. [2]State Key Laboratory of Mechanical Systems and Vibration, School of Mechanical Engineering, Shanghai Jiao Tong University, Shanghai 200240, PR China. [3]These authors contributed equally: Shuzhen Yan, Kaiming Hu. ✉e-mail: wenmingz@sjtu.edu.cn; ponygle@sjtu.edu.cn

controlled wetting of patterned surfaces, whose morphology and superhydrophobicity can be in situ regulated[27]. In particular, wrinkled surfaces resulting from the modulus mismatch between the stiff film and soft substrate in the bilayer system[26,28,29] are a typical example of mechanically unstable surfaces. The research on wrinkled surfaces is important for the development of smart surfaces with dynamically controlled properties because the topological structures of wrinkled surfaces are quite sensitive to external stimuli, including light[30-32], pH[33,34], solvent[35,36], moisture[37,38], and mechanical force[22,39,40]. Photo-driven reactions that can be regulated noninvasively and remotely have already been adopted for controlling wrinkled surface systems by adjusting the crosslink density using intermolecular behavior. For instance, we previously demonstrated the dynamic wrinkled patterns by the regulated modulus distribution resulting from reversible photodimerization of anthracene[41-44] and photocontrolled Diels-Alder reaction[24]. Light-erasable wrinkles operating via softening of the compressive stress field through *trans*-to-*cis* photoisomerization of azobenzene have also been reported[31,45]. Compared to the representative dynamic covalent chemistry, disulfide bonds are known to undergo in situ dynamic non-contacting photo-driven exchange reaction with nearly constant crosslink density[46-51]. In this respect, dynamic wrinkling films provide a promising platform for simultaneously recording molecular behaviors and engineering responsive patterned surfaces.

In this work, based on wrinkled surfaces, we propose a clear and robust strategy for revealing the insightful chemical-induced intermolecular perturbation, macromolecular chain movement, stress relaxation, and microscopic topology (Fig. 1). Upon non-invasive UV irradiation (365 nm), the internal stress relaxation of the crosslinked supramolecular network caused by disulfide exchange can selectively erase the random wrinkled topology in the exposed regions. The variable orientation of wrinkles directly demonstrated the bidirectional interaction between anisotropic mechanical stress fields and polymeric stress relaxation triggered by chemical reactions, which was further confirmed by modified shear-lag theoretical models for one-dimensional (1D) ordered wrinkles. More interestingly, the unexposed wrinkled structures that were initially perpendicular to the exposed boundary presented a seldom orthogonal topographical mutation after heating/cooling (Fig. 1c). The stress-sensitive wrinkle patterns could be repeatedly written/erased and tailored into diversiform topography on demand using the light-induced disulfide exchange reaction. The presented general strategy is not only an important platform for observing other dynamic molecular networks and realizing precise tuning of wrinkles through molecular modulation but also an important step for understanding and thus utilizing infinitesimal perturbations in intermolecular behavior.

## Results

### Strategy for observing the chemistry-induced perturbations

The key strategy for observing the infinitesimal perturbation effects of light-induced disulfide exchange reaction through the dynamic wrinkle patterns is illustrated in Fig. 1. The dynamically crosslinked supramolecular polymer network consisting of pyridine-containing copolymer (PPy-Ba-St, $M_n = 12.1$ kDa, $M_w/M_n = 1.64$) and photosensitive

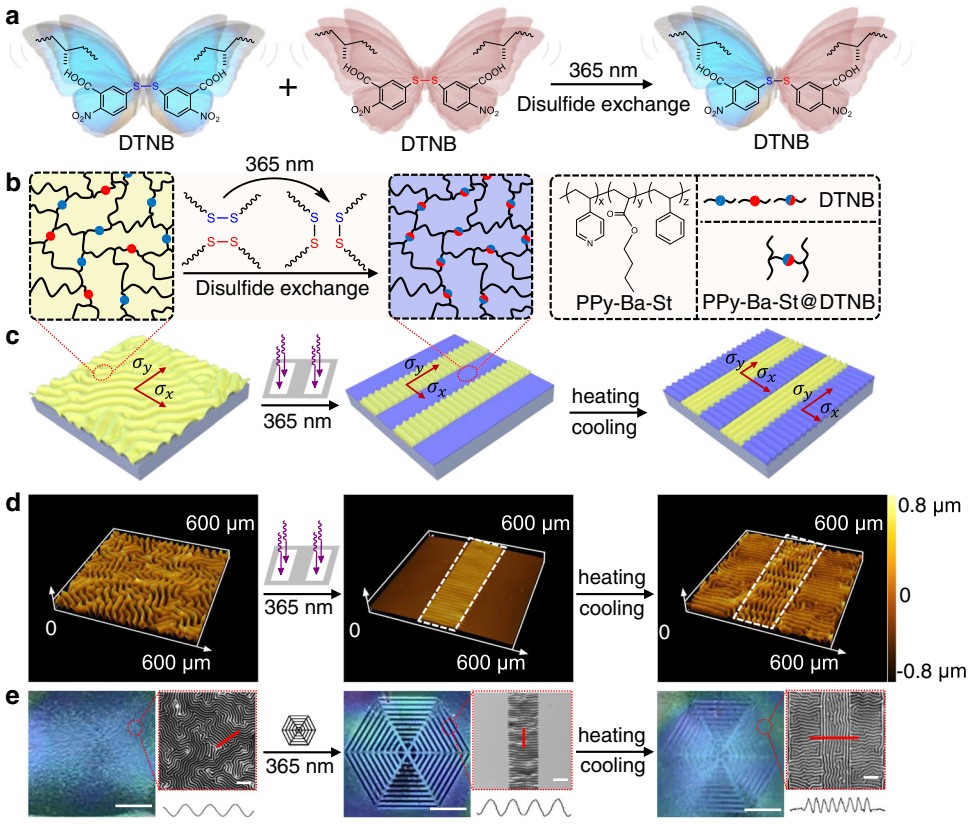

**Fig. 1 | Strategy for observing the infinitesimal perturbation effects of intermolecular behavior through programmable wrinkles. a** The chemical structures and the process of photo-induced disulfide exchange reaction at molecular scale. **b** Schematic representation of dynamic changes of polymer networks according to **a**. **c** Procedures for the fabricating of the tunable self-adaption wrinkle topology based on stress relaxation with a mask induced by 365 nm UV light and heat treatment, the arrows are applied to represent the direction and magnitude of the stress ($\sigma_x$, $\sigma_y$) in different regions. **d** Corresponding laser scanning confocal microscopy (LSCM) images of the microstructures of the top responsive film, the dotted line frames the unexposed region. **e** The optical images of macroscopic patterns with hexagonal mask and heating/cooling treatment (Scale bars: 2 mm), and their partial enlarged detailed 2D LSCM images (Scale bars: 200 μm) and the characteristic tangent profiles of wrinkles corresponding to the marked red lines, the intensity of UV light is 15 mW cm$^{-2}$. Source data are provided as a Source Data file.

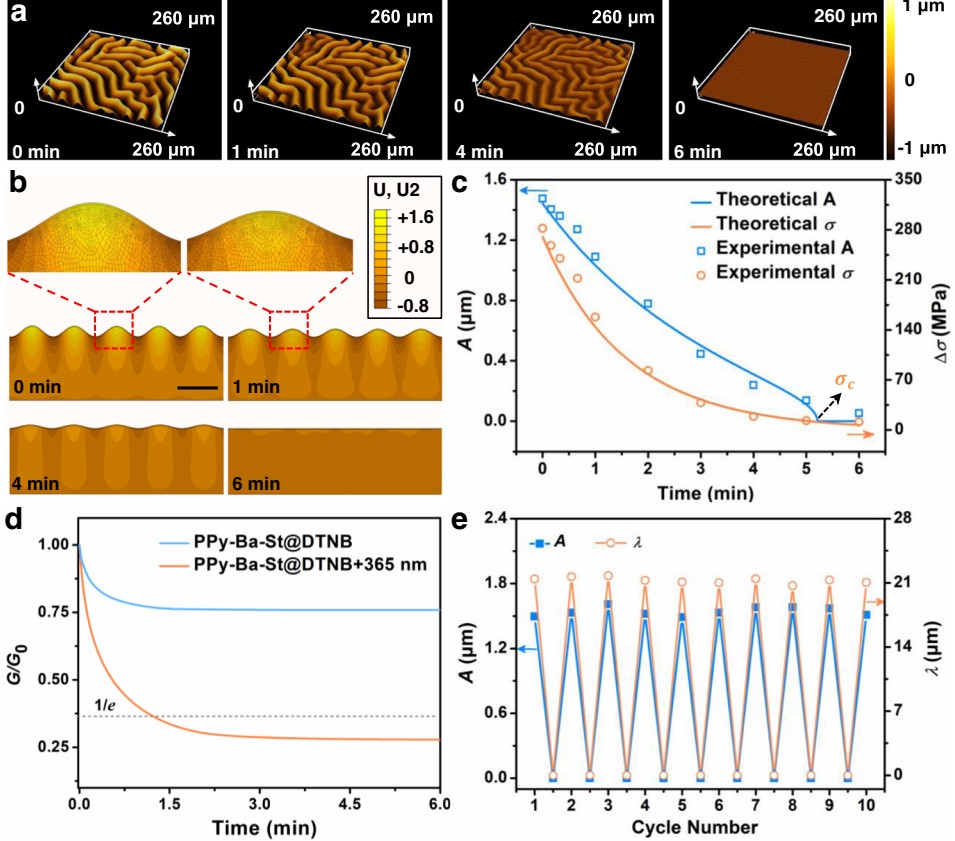

**Fig. 2 | Evolution of the wrinkled morphology by the photo-induced homogeneous stress relaxation of dynamic disulfide exchange reaction. a** 3D LSCM images of time-dependent wrinkle pattern as a function of 365 nm UV light exposure time for 0, 1, 4 and 6 min. **b** FE simulation of stress distributions results corresponding to **a**, and their partial enlarged detailed images were marked by dotted line, "U" and "U2" represent the degrees of freedom and the degrees of freedom in $y$ direction, respectively (scale bar: 20 μm). **c** The experimental $A$, residual thermal stress, theoretical $A$, and residual thermal stress of the wrinkles versus UV irradiation time, $\sigma_c$ represents the critical stress. **d** Stress relaxation curves of PPy-Ba-St@DTNB with and without UV irradiation, $1/e$ represents the stress relaxes to $1/e$ of the initial stress. **e** Generation/erasure cycles of $A$ and $\lambda$ under UV irradiation based on dynamic disulfide exchange reaction. Source data are provided as a Source Data file.

disulfide-containing benzoic acid monomer (DTNB), was spin-coated onto polydimethylsiloxane (PDMS) elastomer as the top layer to form multi-responsive wrinkles based on hydrogen bonding between pyridine and carboxyl groups. The detailed synthesis procedure, characterization of the utilized materials, and the formula of supramolecular system are described in the Supplementary Data (Supplementary Figs. 1–3). The homogeneous labyrinthic wrinkle surfaces with a typically sinusoidal profile emerged on the top stiff film after the bilayer was cooled from 120 °C to room temperature, owing to the mismatch in moduli and thermal expansions coefficient between the top stiff layer and soft PDMS substrate. The process of the photo-driven reaction was recorded by the evolution of disappeared wrinkles. In the case of selective irradiation with 365 nm UV light for 6 min through a strip photomask, the surface wrinkles in the exposed regions were erased by the stress relaxation attributed to intermolecular exchange (Fig. 1a–c), the process of which existed negligible temperature change (Supplementary Fig. 4). At the same time, ordered wrinkles were gradually induced in the unexposed region owing to the in-plane asymmetric distribution of stresses ($\sigma_y > \sigma_x$) generated by the dynamic chemistry.

Upon the heating/cooling treatment (from 120 to 25 °C), the ordered wrinkles first appeared in the exposed region, and subsequently, the horizontal wrinkles located in unexposed regions formed a two-dimensional (2D) orthogonal wrinkle pattern (Fig. 1d) that visualized the infinitesimal perturbations of dynamic disulfide exchange reaction in the exposed region. Furthermore, it is feasible to utilize the infinitesimal perturbations to precisely regulate the

diverse pattern (Fig. 1e and Supplementary Fig. 5). In addition, different patterned surfaces with regional or complex orthogonal wrinkles can visually reveal the exchangeable chemistry as a molecular action detector and also be readily reprocessed based on the photo-induced molecular reaction, which may inspire versatile applications in the fields of information storage and anti-counterfeiting. And this general principle could translate to other systems with appropriate molecular design, such as diselenide-containing molecules.

## Photo-induced stress relaxation and wrinkle erasure

The effect of dynamic disulfide exchange reaction on the evolution of wrinkle topography was investigated by LSCM. As shown in Fig. 2a, the random wrinkling structures gradually disappear with increasing irradiation time: the corresponding experimental amplitude ($A$) of the wrinkles decreases from 1476 to 56 nm upon UV irradiation for 6 min, which agrees well with the theoretical results (Fig. 2c). At the same time, wavelength ($\lambda$) does not change significantly, indicating that no significant changes occurred in the modulus of the supramolecular network (Supplementary Fig. 6). Finite element (FE) simulations were used to model the erasable wrinkles and stress distributions in the top supramolecular network after intermolecular exchange induced by 365 nm irradiation (Fig. 2b). As shown in Fig. 2c and Supplementary Fig. 7, the experimental residual thermal stress is consistent with the theoretical results. The residual thermal stress after the exposure time of $t > 5.2$ min is lower than the critical stress triggering wrinkle instability ($\triangle\sigma < \sigma_c$), therefore, the wrinkles can be erased through

photo-induced stress relaxation, on the other hand, the stress relaxation was visualized by the time-dependent erasure of the wrinkle patterns (Fig. 2a, c).

To further elucidate the mechanism of the erasable wrinkles induced by photo-induced exchange reaction, we further relaxed the internal stress in the polymer network. As shown in the dynamic thermomechanical analysis curve (Fig. 2d), the stress in the PPy-Ba-St@DTNB specimen significantly decreases after sequential exposure to UV light. Notably, the relation of time-varying macroscopic stress in the top film to molecular scale chain length change induced by photo-driven disulfide exchange can be expressed as:

$$\triangle\sigma(t) = kT_aN\nu(\frac{r_0}{Nb})^2[\theta^2(t) - 1/\theta^2(t)] \tag{1}$$

where $k$, $T_a$, $N$, and $\nu$ are the Boltzmann constant, absolute temperature, number of monomers in a chain segment, and density of chains (number of chains per unit area); $r_0$, $b$, and $\theta(t)$ are the average magnitude of end-to-end vectors in the un-deformed state, Kuhn monomer length, and the change in compression ratio with exposure time, respectively. Equation (1) shows that the macroscopic mechanical properties of wrinkled surfaces can be well depicted by the time-varying molecular scale chemical reaction (the details shown in Supplementary Note 1). In addition, the wrinkle morphology is highly reversible, for at least tens of irradiation cycles by virtue of the stress relaxation caused by the photodynamic disulfide exchange reaction and heating/cooling treatment, suggesting remarkable reprocessing performance (Fig. 2e).

To investigate the role of photodynamic disulfide bonds in the actuation of the wrinkle patterns, we prepared three specimens with different top films; the detailed comparison is presented in Supplementary Table 1. The sample 1 only contains pyridine-containing copolymer PPy-Ba-St, sample 2 is comprised by polymer PPy-Ba-St and disulfide-containing molecule (Phenyl disulfide). In addition, the sample 3 (PPy-Ba-St@H2MDB) and sample 4 (PPy-Ba-St@DTNB) are a supramolecular polymer network through the hydrogen bonding between pyridine and carboxyl groups. As seen in a series of LSCM images (Supplementary Fig. 8), the wrinkles can be generated at the surface of PPy-Ba-St@H2MDB (sample 3) and PPy-Ba-St@DTNB (sample 4) system owing to the thermal compressive stress exceeds the critical stress, while the wrinkle patterns of the sample 3 remained consistent regardless of whether it was irradiated by the UV light because of no dynamic disulfide exchange reaction. However, when we use other polymer blends PPy-Ba-St (sample 1) and PPy-Ba-St@Phenyl disulfide (sample 2) as top film, no pattern is produced on the surface owing to the lack of crosslinking by hydrogen bonds between pyridine and carboxyl groups. These results confirm that the crosslinked hydrogen bonding is a key driver of wrinkles' generation, as well as the photodynamic disulfide exchange reaction which makes wrinkle patterns reversible and responsive to light irradiation. Here, the disulfide exchange reaction is assumed to go through a radical mechanism[51,52]. The electron paramagnetic resonance (EPR) was used to verify the generation of thiol radical in the presence of the radical scavenger 5,5-dimethyl-1-pyrroline N-oxide (DMPO). As shown in Supplementary Fig. 9, the signal of a thiol radical was detected after UV light irradiation. Additionally, leucocrystal violet (LCV) was used to confirm the radical generation directly (Supplementary Fig. 10). Upon UV irradiation, the colorless toluene solution of DTNB changed its color to bright purple because of the chemical reaction between the LCV and thiol radicals. These results indicate that the photoinduced disulfide exchange in the PPy-Ba-St@DTNB supramolecular crosslinked network, which contributes to the stress relaxation in the wrinkled top layer, is based on the coupling reaction of thiol radicals.

## Abrupt transition of wrinkling orientation

To verify the stress relaxation induced by photochemical disulfide exchange reaction can be visualized through ordered wrinkles, we reconfigured the stress distributions and mechanical boundary conditions by selective irradiation with a photomask for the system of PPy-Ba-St@DTNB. Thanks to the non-invasive and high spatiotemporal stimulus provided by UV light, the ordered wrinkles were subsequently generated. As shown in Fig. 3a and Supplementary Fig. 11, the labyrinthic wrinkles in the exposed regions are selectively erased after exposure to 365 nm UV light through a striped photomask. Noteworthy, the initially disordered topological structures in the unexposed region, upon UV light irradiation, gradually evolved into highly ordered wrinkles that are perpendicular to the light-defined boundaries, without significant change in $\lambda$. The erasure of wrinkle is attributed to the stress relaxation by disulfide exchange reaction in exposed regions. For the unexposed regions, owing to the boundary constraint effect[25,28], the wrinkled pattern is going to rearrange to adapt itself to the light-defined boundary (the details shown in Supplementary Discussion Note 2), and the corresponding theoretical models is shown in Fig. 3g. Then, we treated it with heating/cooling, surprisingly, the ordered wrinkles first appeared in the exposed region because of the infinitesimal perturbations of disulfide exchange reaction, while the original perpendicular wrinkles in the unexposed region mutated to parallel to the light-defined boundary. The evolution of orthogonal wrinkle patterns was further confirmed by the light diffraction patterns inset because of the microstructure-affected light transmission path. The initially disordered topological structures presented a light diffraction pattern with concentric rings (Supplementary Fig. 12), which were viewed by a laser beam (650 nm). After a selective erasure with a striped photomask upon 365 nm UV light, the distinguishing discrete diffraction points were obtained. When the laser passed through the samples without topological structures after heating/cooling treatment, one laser point can be observed on the receiving plane first, and the rectangular diffraction patterns were observed subsequently, which verified the generation of orthogonal wrinkle patterns.

We then conducted a series of controlled experiments to confirm the trans-scale mechanical mechanism of these phenomena. First, the $E_f$ of the top layer was determined by tapping-mode AFM (Supplementary Fig. 13), which showed no significant fluctuations in $E_f$ after UV light irradiation. Additionally, the transformation of DTNB after irradiation with 365 nm light was not detected using UV–visible spectroscopy (UV–vis), Fourier transform infrared spectroscopy, and [1]H NMR spectra (Supplementary Fig. 14), which excluded the influence of photoinduced side-reactions. These results demonstrate that infinitesimal perturbations of the disulfide exchange reaction at molecular scale lead to macroscopic temporal-spatial transformation in wrinkle topography, especially, abrupt change in wrinkling directions.

Although it was confirmed that wrinkling morphology can be changed via irradiation with 365 nm UV light, the stress distribution of the supramolecular polymer surface remains ambiguous and should be explored deeply. Hence, a 4-chain constitutive model of photosensitive polymer networks in the exposed regions was proposed (Fig. 3b–f) for establishing the relationships between the intermolecular behavior, structure of polymer networks, and their mechanical properties of macroscopic polymer films. In this model, the polymer networks are assumed to be assemblies of nonlinear springs connected at crosslinking points. Before UV light exposure, the polymer chains (Fig. 3b) are subjected to the macroscopic isotropic thermal compressive stress $\sigma_0$. The principal compression ratios of polymer chains along the x-axis and y-axis directions in deformed random networks are defined as $\theta_1$ and $\theta_2$, respectively, and the average length of polymer chains in un-deformed state is defined as $r_0$. After UV exposure, the photo-induced disulfide exchange reaction can relax the deformed polymer chains to their initial equilibrium positions, which is similar to the restoration of the compressed spring to its

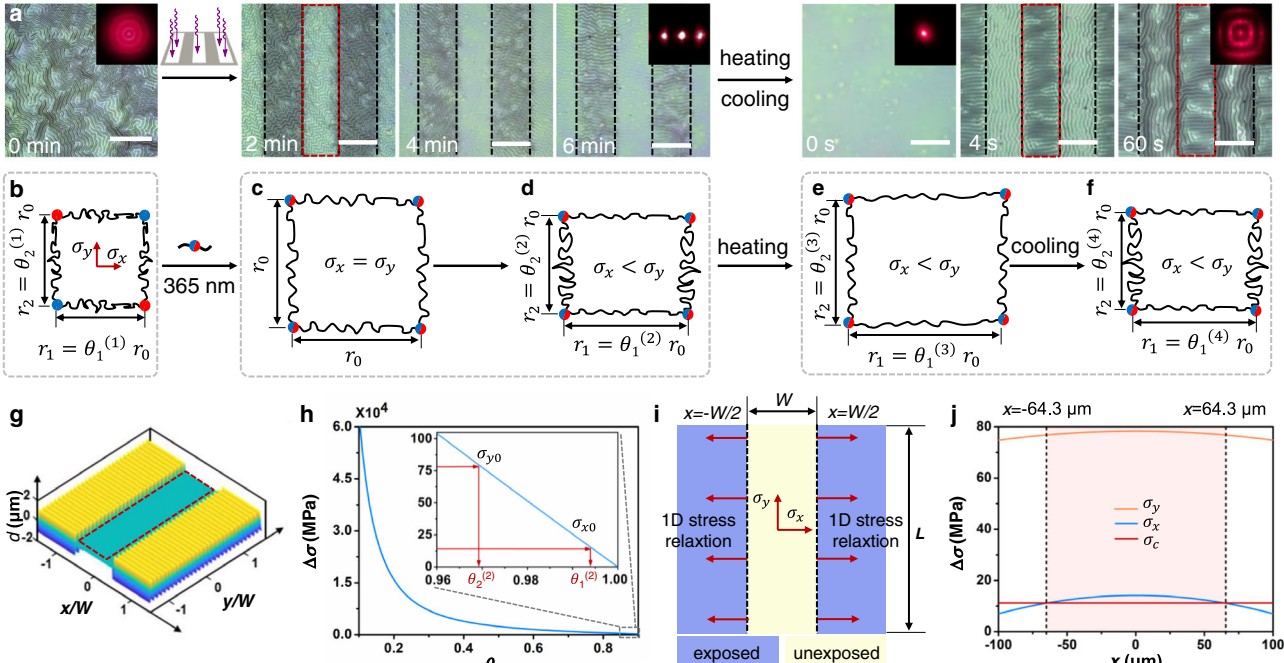

**Fig. 3 | Trans-scale mechanical mechanism of the micro-wrinkle patterns with abrupt orientation transition perturbed by photochemical reaction at molecular scale. a** Optical images of the morphological changes in exposed and unexposed regions when the samples of labyrinth wrinkle patterns were exposed and heated/cooled for different times under a strip mask. The inset pictures are the corresponding light diffraction patterns. Scale bars: 200 μm. 4-chain constitutive model of the photosensitive polymer networks (PPy-Ba-St@DTNB): **b** before UV light exposure; **c** isotropic molecular chain stretching under free state without boundary constraints and **d** anisotropic molecular chain stretching under constraint state by boundary stresses after UV light exposure; **e** thermal expansion and **f** cold contraction. **g** The theoretical contours of 1D ordered wrinkles perpendicular to the exposure boundary after UV light exposure. **h** Stress differences with respect to the compression ratio θ in exposed regions. **i** The modeling of 1D stress relaxation of the unexposed region is triggered by the disulfide bond exchange reaction of the exposed region. **j** Stress distributions in the unexposed region. The dotted box represents the exposed region. Source data are provided as a Source Data file.

equilibrium point. In this process, the stress relaxation can be illustrated using two-step modeling, including isotropic molecular chain stretching under free states without boundary constraints (Fig. 3c) and anisotropic molecular chain stretching under the constrained state by boundary stresses (Fig. 3d). In the second step, the disulfide exchange-induced asymmetrical 2D stress relaxation is limited by the boundary stresses. According to Eq. (1), the boundary constraint stresses $\sigma_{x0}, \sigma_{y0}$ and the principal compression ratios $\theta_1^{(2)}, \theta_2^{(2)}$ along the 1st and 2ed directions satisfy the following relationship:

$$|\sigma_{x0}| = -kT_aN\nu\left(\frac{r_0}{Nb}\right)^2[\theta_1^2 - 1/\theta_1^2] \tag{2a}$$

$$|\sigma_{y0}| = -kT_aN\nu\left(\frac{r_0}{Nb}\right)^2[\theta_2^2 - 1/\theta_2^2] \tag{2b}$$

According to Eq. (2) and Fig. 3h, when $1 > \theta_{(t)} > 0$, the stress differences monotonically decrease with the compression ratios $\theta_{(t)}$. Because of the stress relaxation at the boundaries of the unexposed area, the stresses at the boundaries ($x = W/2$ or $x = 3W/2$) (Supplementary Fig. 15), follow the order of $\sigma_{x0} < \sigma_{y0} < \sigma_0$ ($\sigma_{x0} = 14.17$ MPa, $\sigma_{y0} = 78.33$ MPa). Furthermore, we obtained that $1 > \theta_1^{(2)} > \theta_2^{(2)}$, ($\theta_1^{(2)} = 0.9945$, $\theta_2^{(2)} = 0.9697$), which indicates that the polymer network under the constrained boundary stress can be slightly compressed after photo-induced disulfide exchange. Moreover, according to Supplementary Eq. (5) and Supplementary Fig. 16, $E_0 < E_x < E_y$, which indicates that the heterogeneous surface films are produced via disulfide bond exchange-induced asymmetrical stress relaxation, which leads to a very small increase in the elastic modulus of the surface film. The increment of elastic modulus is too small to be characterized (Supplementary Fig. 13). However, the minor change of elastic

modulus induced by infinitesimal perturbations in the intermolecular exchange reaction can trigger the macroscopic wrinkle morphological mutations, which just illustrated that the insignificant perturbations can result in chain reactions in macroscopic systems.

For the unexposed area, owing to the boundary constraints of the exposed region, the 1D stress relaxation occurred at the boundaries ($x = \pm W/2$) (Fig. 3i). As indicated in Fig. 3j and Supplementary Eq. (17), the 1D relaxation along the $x$ direction can cause non-uniform stress distributions. When $x < -64.3$ μm or $x > 64.3$ μm, $\sigma_y > \sigma_c > \sigma_x$, indicating that the 1D ordered wrinkles along the $y$ direction will be triggered before the wrinkles along the $x$ direction (Fig. 3j). Therefore, the initially random wrinkles in the unexposed regions dynamically evolved into highly ordered wrinkles after irradiation with UV light for 6 min (Fig. 3a).

When the samples are heated, in the exposed regions, the polymer chains along the 1st and 2ed directions in deformed random networks are stretched (Fig. 3e), and the average lengths can be given as $r_1 = \theta_1^{(3)}r_0$ and $r_2 = \theta_2^{(3)}r_0$. The average lengths during cooling (Fig. 3f) can be given as $r_1 = \theta_1^{(4)}r_0$ and $r_2 = \theta_2^{(4)}r_0$ (the details shown in Supplementary Discussion Note 3). Because of the thermal expansion of PDMS and the heterogeneous boundaries along the $x$ and $y$ directions in the exposed region, we can obtain $\theta_1^{(4)} = \theta_1^{(2)} > \theta_2^{(4)} = \theta_2^{(2)}$. The elastic moduli of the polymer network satisfy the following relationship $E_0 < E_x < E_y$. According to $\varepsilon^c = (3\bar{E}_s/\bar{E}_f)^{2/3}/4$ ($\bar{E}_f$ and $\bar{E}_s$ refer to the plane strain moduli of the surface film and substrate, respectively), the critical strain $\varepsilon_x^c$ is larger than the critical strain $\varepsilon_y^c$, which indicates that the wrinkles along the $y$ direction can be first triggered before the wrinkles along the $x$ direction in the exposed area. For the unexposed region, the elastic modulus of surface film is slightly smaller than the hard constraint boundary of the exposed region, $E_{ue} = E_0 < E_e = E_x$. Therefore, the stresses in the unexposed surface film satisfy the

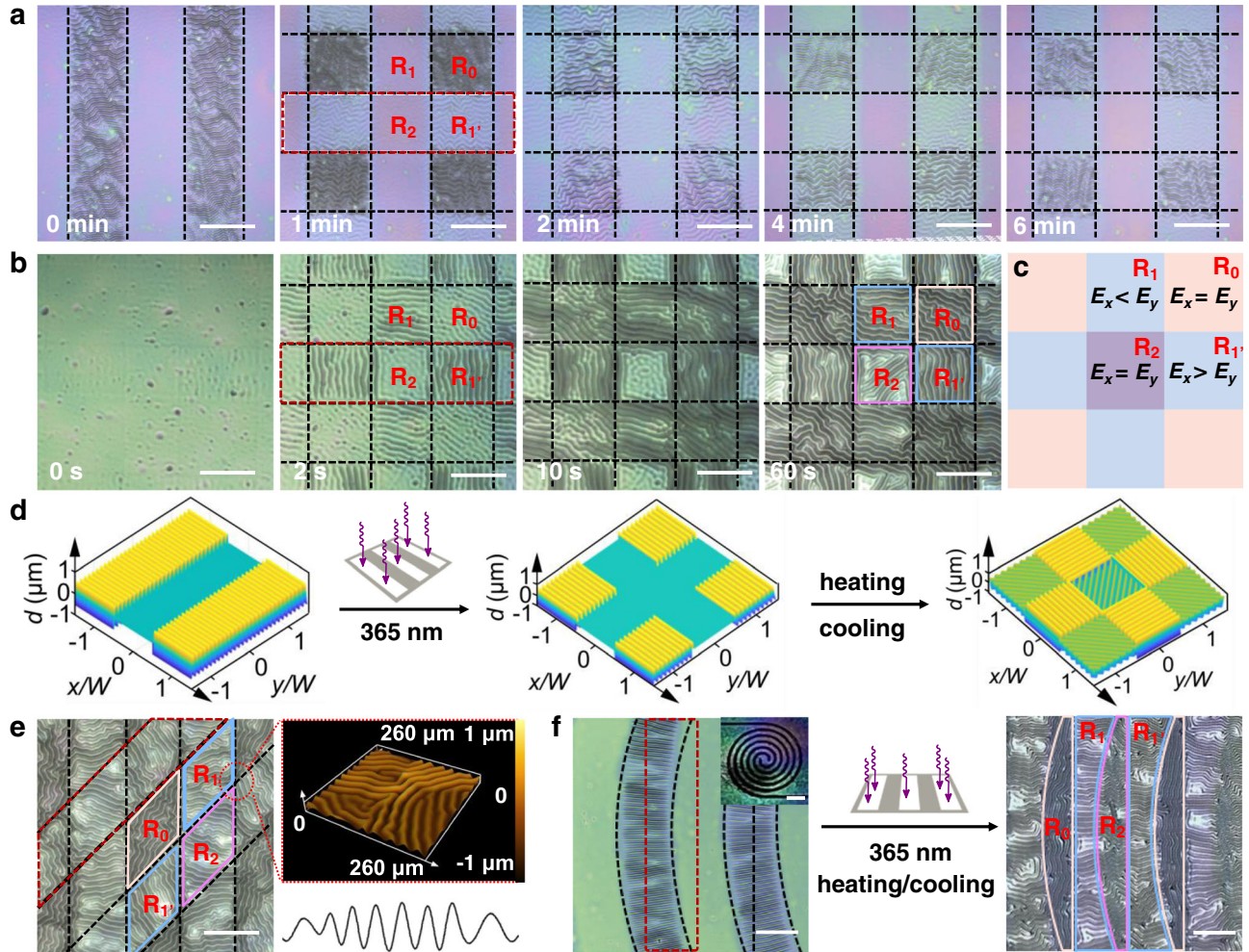

**Fig. 4 | Sequential exposure strategy for tuning alternative wrinkle micro-structures. a** Evolution process of the wrinkle morphology by selectively irradiation for 0, 1, 2, 4 and 6 min through the strip photomask rotated by 90°.
**b** Formation process of the wrinkle morphology with variable orientation by heating and cooling treatment for 0, 2, 10 and 60 s. **c** The mechanical model of a repeatable unit including domains $R_0$–$R_2$ in the top film. **d** Schematic illustrations of the formation of wrinkles via sequential irradiation. **e** Regulation of 2D ordered wrinkle morphologies during exposure at the angle of 45°; the insets are the corresponding 3D LSCM image and the characteristic tangent profile of the regulated wrinkle corresponding to the marked line. **f** Optical images of the wrinkle morphology with helicoid pattern under the sequential exposure strategy with strip photomask. The inset pictures are the corresponding large-scale patterns (Scale bar: 1 mm). The dotted box represents the exposed region. Scale bars: 200 µm. Source data are provided as a Source Data file.

following relationship: $\sigma_x > \sigma_c > \sigma_y$, resulting in the formation of wrinkles parallel to the boundaries ($x = \pm W/2$) in the unexposed area (Fig. 3a). Thus, the microscopic wrinkling orientation visually differentiates the disulfide exchange reaction in the top film.

## Sequential spatiotemporal perturbations

To further reveal the mechanical mechanism of the photo-programmable chemical reaction and internal stress for the above phenomenon, we fabricated diverse wrinkle patterns via sequential UV light exposure. As shown in Fig. 4a, the 1D ordered wrinkle patterns in area $R_{1'}$ are erased after the second exposure for 6 min through the strip photomask rotated by 90°. After heating and cooling, the wrinkle morphology with variable orientations formed at the surface (Fig. 4b). As shown in Fig. 4c, $R_0$, $R_1$, $R_{1'}$, and $R_2$ are the unexposed, first-exposed, second-exposed, and double-exposed areas, respectively. Owing to the double physical confinement effect of sequential exposure[25], the elastic modulus along the $x$ direction is equal to the elastic modulus along the $y$ direction ($E_x = E_y$) in $R_0$ and $R_2$. For heterogeneous surface film in the single exposed area, the elastic moduli of the polymer network satisfy the following relationship $E_x < E_y$ ($R_1$) and $E_x > E_y$ ($R_{1'}$);

therefore, a discontinuous orthogonal wrinkle pattern is obtained. The corresponding theoretical models of the wrinkle patterns generated by the three different processing strategies are shown in Fig. 4d.

Furthermore, we adjusted the exposure angle to 45°, consequently, diverse wrinkles on the surface can be available with different wrinkle morphologies and orientations (Fig. 4e and Supplementary Fig. 17); the detailed evolution of the wrinkle morphology is given in Supplementary Fig. 18. To verify the feasibility of the sequential exposure strategy, we replaced the strip photomask with a helicoid photomask at first exposure (Fig. 4f). The morphological evolution of the resulting surface is consistent with that obtained using orthogonal exposure. These results show that the transition of the wrinkle orientations is strongly dependent on the disulfide bond exchange-induced stress relaxation during selective irradiation. In addition, the wrinkle shows different physical properties (morphology and orientation), which can be used to visualize the perturbation effect of photochemical intermolecular behavior in the differently exposed regions.

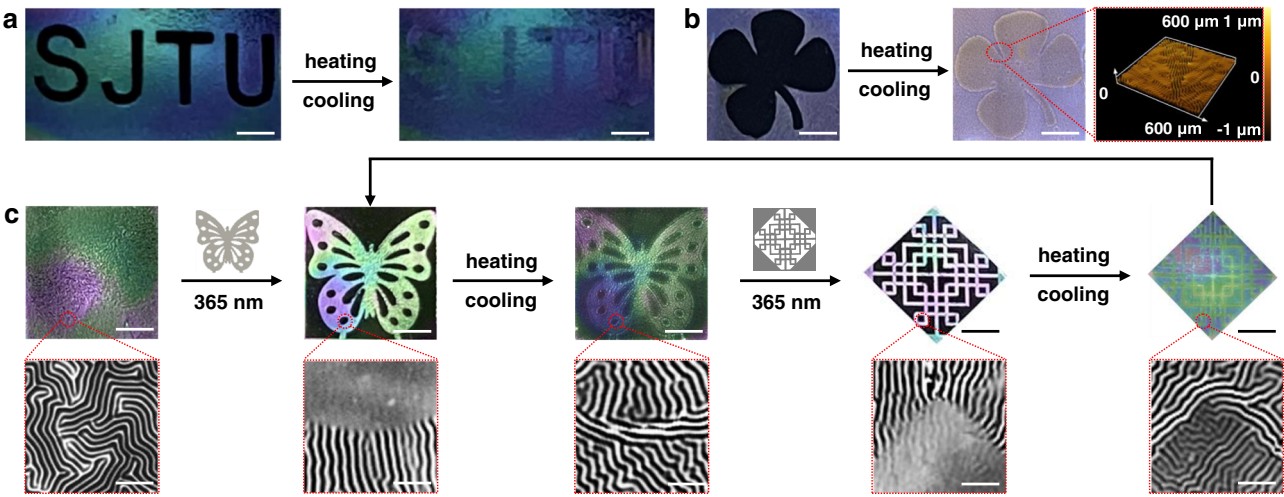

**Fig. 5 | Wrinkle patterns with spatial features formed using perturbation effect of dynamic disulfide exchange reaction.** Optical images of the PPy-Ba-St@DTNB film was selectively exposed through the mask with **a** letter "SJTU" and **b** "clover" shape, respectively, then treated with heating/cooling treatment; the insets are LSCM images of the orthogonal wrinkle on the boundary, Scale bars: 1 mm; **c** The optical images (top, Scale bars: 1 mm) and heir partial enlarged 2D LSCM images (bottom, Scale bars: 100 μm) of the reversible pattern from the butterfly pattern to the Chinese knot pattern.

### Reprogrammable patterns based on molecular editing

Utilizing the perturbation effect based on photo-induced disulfide exchange reaction in the top film, we can selectively eliminate the stress and prepare a series of highly predetermined wrinkle patterns through simple selective exposures (Figs. 2 and 3). The reversible wrinkle patterns might be suitable for developing next-generation smart materials that can facilely adjust their surface geometry and properties in response to external stimuli. A series of diverse patterns such as "letter", "clover", "butterfly", and even "Chinese knot" could be formed on the surface of the bilayer material. As shown in Fig. 5a, b, letters "SJTU" and the clover shape were photolithographed on the high-definition surface and subsequently selectively erased. After heating and cooling treatment, the patterns were converted into a fully wrinkled surface with orthogonal wrinkles at the edge of "SJTU" and "clover", which was considered writing/encryption of information. Furthermore, we produced the butterfly pattern on the wrinkled surfaces, which precisely corresponded to the used photomask, and the original random wrinkles were turned to parallel to the light-defined boundary (Fig. 5c). We then heated this sample at 80 °C, which erased the butterfly pattern, and cooled it to room temperature, which made the entire surface wrinkled with an orthogonal wrinkle pattern on the boundary of the butterfly pattern (Fig. 5c). Then, the bilayer was irradiated with UV light through a photomask to produce a "Chinese knot" pattern. In these demonstrations, wrinkles acted as a molecular action detector, revealing the perturbation effects in the photodynamic disulfide exchange reaction upon UV irradiation. At the same time, the complicated patterning information could be iteratively written onto the surface of the bilayer and made visible to the naked eye because of the differences in light scattering intensity between the wrinkled and erased regions. This approach may be used for smart information displays, dynamic no-ink printing, and anti-counterfeiting.

Aside from irradiation with UV light, the wrinkles could also be eliminated using acids owing to the hydrogen bonds between pyridine groups in PPy-Ba-St and carboxyl groups in DTNB. As shown in LSCM images (Supplementary Fig. 19a), during the treatment with HCl gas, $A$ rapidly decreases with the treatment time while $\lambda$ remains almost unchanged. The characteristic $A$ decreases from 1500 to 128 nm after acid treatment for 25 s, revealing that the investigated bilayer material is responsive to pH changes (Supplementary Fig. 19b). After the film was heated to evaporate HCl, the supramolecular network became crosslinked again so that the wrinkle patterns were recovered. As shown in Supplementary Fig. 14c, the letter "Y" or "Z" was written with 5 wt% HCl using a writing brush. In the written area, the wrinkles flattened and were transparent, so that the letter "Y" or "Z" could be distinguished with the naked eye. Various patterns can be realized using this simple approach (Supplementary Fig. 20), demonstrating that this method is versatile and can be applied in smart displays. These wrinkle patterns can be applied to the indication of goods storage and transportation conditions, which is sensitive to the changes of conditions, such as heating, UV lighting, and pH. For instance, owing to the rarely transformable orientation of microstructures, the wrinkles can be used as a special anti-counterfeiting label to monitor whether the medicines have experienced deactivated temperature during transport (Supplementary Fig. 21).

### Discussion

We demonstrated a facile and robust strategy for visualizing intermolecular perturbation behavior based on the photodynamic chemical reaction within supramolecular network through the changing macroscopic surface patterns. As an initial presentation of intermolecular perturbation, the sensitive wrinkling topology revealed the localized stress relaxation in supramolecular film owing to UV light-induced disulfide exchange reaction, which was quantitatively described for inspiring the progress of dynamic chemistry and material mechanics. Moreover, the experimental results and transscale theoretical analysis concurrently indicated that selective molecular regulation enables the evolution of well-defined ordered microstructures from random wrinkles. These patterns could be mutated to perpendicular through the perturbation of the as-wrinkled stress field. This effect can be applied for eliminating the undesired surface wrinkles and also fabricating patterned surfaces, which strongly realized unprecedented applications in the molecular behaviors of dynamic chemistry. Owing to the photo-modulated supramolecular layer, the multitudinous information can be repetitively written and erased on the surface. We envision that the presented strategy using responsive wrinkles as a molecular action detector will deepen a more insightful understanding of the dynamic chemical system in the future. And the photo-regulated molecular editing is promising to facilitate the design and development of smart surfaces.

## Methods

### Materials

Styrene (99%) and *n*-butyl acrylate (BA) (99.5%) were obtained from China National Pharmaceutical Group. 5,5′-Dithiobis-(2-nitrobenzoic acid) (DTNB) (99.9%) was obtained from Energy Chemical (Shanghai, China). 5,5-dimethyl-1-pyrroline N-oxide (DMPO) (99%) was obtained from Sinopharm Chemical Reagent Co. Ltd. (Beijing, China). PDMS was provided by Dow Corning Inc. (Michigan, America). All other chemicals were provided by Adamas-Beta Co. Ltd. (Shanghai, China).

### Characterizations

The chemical structures of the synthesized compounds (8 mg ml$^{-1}$) were verified by $^1$H nuclear magnetic resonance ($^1$H NMR) using a 400 MHz Advance NMR spectrometer (Bruker, Germany) at room temperature. Average molecular weights of the obtained copolymers (2 mg ml$^{-1}$) were determined by means of gel permeation chromatography (GPC, LC-20A, Shimadzu, Japan), using tetrahydrofuran (THF) as an eluent at a flow rate of 1.0 ml min$^{-1}$ with a combination of two columns (Shodex, KF-802 and 804, 300 × 8 mm) and equipped with a RID-10A differential refractive index detector. EPR spectra were recorded on an EPR spectrometer (13000 Gauss, Bruker, Germany). And the AFM images were conducted by utilizing a scanning probe microscope (Dimension Icon & Fastscan, Bruker, Germany), which operated in tapping mode with silicon cantilevers (with a force constant of 40 Nm$^{-1}$). UV–vis absorbance spectra were detected on a Shimadzu UV-3390 spectrophotometer (Hitachi, Japan). Furthermore, the profile measurement microscope (VF-7510, KEYNCE, Japan) and LSCM (LEXT OLS5000, Olympus, Japan) were utilized to observe the dynamic wrinkling structures.

### Synthesis of pyridine-containing copolymer (PPy-Ba-St)

The pyridine-containing polymer PPy-Ba-St was synthesized via free-radical copolymerization according to Supplementary Fig. 1. In detail, 4-vinylpyride (3.70 g, 36 mmol), *n*-butyl acrylate (9 g, 72 mmol), and styrene (8.32 g, 72 mmol) were dissolved in 30 ml 1,4-dioxane at a feed molar ratio of 1:2:2. Subsequently, 200 mg 2,2-azobisisobutyronitrile was added (1% of the total monomer weight). The polymerization reaction was performed at 70 °C for 12 h under nitrogen protection. The reaction mixture was precipitated three times in cold petroleum ether. After the mixture was filtered, the product was dried at 70 °C for 24 h, yielding the solid copolymer. The structure of PPy-Ba-St was verified by $^1$H NMR spectrum as shown in Supplementary Fig. 2, the assignment of the proton peaks in the $^1$H NMR spectrum demonstrated that the molar ratio of pyridine, butyl acrylate, and styrene was -1:1.6:2.4. $^1$H NMR (500 MHz, CDCl$_3$, ppm): $\delta$ = 8.49–8.01 (CH-N-CH), 7.25–6.12 (benzene or pyridine ring), 4.18–3.25 (-O-CH$_2$-), 2.41–1.03 (-CH-CH$_2$-), 1.01–0.78 (-CH$_3$).

### Preparation of PDMS elastic substrates

The PDMS elastic sheet substrate was prepared by mixing PDMS prepolymer (Sylgard 184, Dow Corning) at a 15:1 base/curing agent ratio, followed by being drop-coated in a Petri dish, degassing degassed in a vacuum oven, and cured at 70 °C for 4 h (thickness: ~400 μm). Then, the sample (thickness: ~400 μm) was cut into 1 cm × 1 cm and or 1.5 cm × 1.5 cm squares, respectively.

### Preparation of photosensitive disulfide-containing wrinkle pattern

We composited PPy-Ba-St and DTNB with different molar ratios for pattern preparation and observation of perturbation behavior. As shown in Supplementary Fig. 3, too little DTNB will directly lead to the inability to form a wrinkled pattern on the surfaces (PPy-Ba-St: DTNB = 10:0/10:0.5). As the concentration of DTNB increased, phase separation phenomenon can be observed with inhomogeneous distribution of wrinkled pattern on the surface induced by the excessive

crosslinking intensity between pyridine and carboxyl groups. In this work, we employ the molar ratio of 10:1 (PPy-Ba-St: DTNB = 10:1) to carry out subsequent experimental studies.

### Preparation and erasure of photosensitive wrinkle pattern

A THF solution of PPy-Ba-St (3 wt%) and DTNB with different molar ratios was filtered through a 0.22 μm filter, and spin-coated onto the prepared PDMS elastic substrates to prepare the multi-functional bilayer systems, which can realize the visualization of photo-induced stress relaxation by dynamic wrinkles. The bilayer samples were then heated at 80 °C for 3 min. After cooling to room temperature, the wrinkled patterns generated. The wrinkled samples could be erased by 365 nm UV light or HCl gas.

### FE analysis of the stress distribution of the bilayer

FE analysis of the stress distribution of the bilayer wrinkled surfaces subjected to an increased UV irradiation time was performed in the static environment of the Abaqus software (version 2020). The PPy-Ba-St@DTNB and soft PDMS bilayered system were constructed from top to bottom according to the experiments as shown in Fig. 2b, and the interfaces among them were assumed to be bonded without slippage. As for the boundary conditions, the bottom of the bulk material was constrained in the y direction, and the PDMS elastic substrate could move only along the indentation direction when subjected to a linearly increasing displacement load. The linear elastic model was utilized to qualitatively describe the constitutive behavior of the bulk materials.

Details of material analysis can be found in the Supplementary Information.

### Reporting summary

Further information on research design is available in the Nature Portfolio Reporting Summary linked to this article.

## Data availability

The authors declare that the data supporting the findings of this study are available within the paper and its Supplementary files. The data that support the findings of this study are available from the corresponding authors upon request. Source data are provided with this paper.

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

## Acknowledgements

The authors thank the National Key R&D Program of China (2021YFB4001100), and National Nature Science Foundation of China (52025032, 12032015, and 12172216) for their financial support.

## Author contributions

S.Y. and X.J. conceived the concept and designed the experiments. S.Y. and S.C. characterized the materials, performed the measurements and wrote the manuscript. S.Y. and K.H. carried out the mechanical analysis. X.J., W.Z., J.Y. and T.L. supervised the research. All authors contributed to the work.

## Competing interests

The authors declare no competing interests.
