## [Peer Review File · Nature Communications]

Reviewer comments, first round –

Reviewer #1 (Remarks to the Author):

In the article by Jiang, Zhang and colleagues under consideration at Nature Communications, the authors report a method to observe dynamic chemistry at interfaces using light. Central to this idea is the interaction of disulfide molecule DTNB that integrates with a PPy-Ba-St polymer film through hydrogel bonding –UV irradiation triggers wrinkles in the film accompanied by disulfide exchange reactions. The authors reflect on how molecular changes have a “butterfly effect” leading to macroscopic phenomena. Intriguingly, the reactions at interfaces are guided by stress relaxation and the gradient properties of the network is shown to propagate after heating/cooling with scope for using this approach to encode information within supramolecular films. This is an exciting work that is carefully done with nice theory, computation and experiment supporting the proposed mechanisms. I believe it warrants publication after considering some minor points as follows:

The presumed hydrogen bonding between the carboxylate of the DTNB and N of the pyridine are proposed to be key drivers in wrinkle generation. However, there is little evidence that H-bonding is a central mechanism. Hydrogen bonding will also be influenced by temperature changes –could changes in H-bonds after illumination influence molecule mobility and yield the wrinkling orientation behaviour? The incorporation of DTNB into the film needs to be demonstrated through supporting experiments (e.g. FTIR/Raman stretching modes with H-bonds, etc.) and discussed in more detail to guide the reader in the molecular characteristics. For instance, what is the concentration/density of DTNP in the polymer and how does this effects the observed behaviour?

The demonstrations in the latter half of the paper are compelling. One addition that would broaden the scope of the work for the journals broad audience would be some discussion of how these concepts and experiments might be translated to other materials systems. Is this behaviour unique to this system or is there some general principles in macromolecular design that could translate to other systems? Some discussion of how other dynamic bonds, polymer systems with different viscoelastic properties, etc., might be amenable to this approach would be beneficial. If this phenomenon is highly specific to this system, this should be stated with some examples of how future studies will yield new knowledge, new devices, etc.

On page 8, line 184 experiments are described using control materials. It is not clear in the main manuscript what the (i) and (ii) materials are without referring to SI. These should be described better in the main text with specific mention to the SI.

The light diffraction patterns shown as insets of Figure 3 are not well explained. More text is needed to explain how this result supports the orientation transition.

Reviewer #2 (Remarks to the Author):

This paper is clearly incremental research compared with papers published from the same group, such as: “Acc. Chem. Res. 2019, 52, 4, 1025–1035”, “Science Advances 2018, 4 (4), eaar5762”, “Adv. Mater. 2020, 32, 1906712”, and “Nat. Commun. 2020, 11 (1), 1811”. Especially, the mechanical model has been shown in “Adv. Mater. 2020, 32, 1906712”(Figure 3). The authors didn’t review these papers properly and comment the difference between this paper, some of which were not even cited. The authors claim that “wrinkle patterns could be repeatedly written/erased”. But this phenomenon has been reported in several papers. Generally, this paper shows a bit of improvement of the “wrinkle pattern” module. I think it won’t attract common interest. Ant the novelty of this paper doesn’t reach the requirement of “Nature Communication”. Thus, I suggest to reject this paper.

Reviewer #3 (Remarks to the Author):

This article constructs a supramolecular cross-linked polymer network, which can undergo rearrangement through UV-induced disulfide bond exchange. A dynamic wrinkled pattern based on the material allows the visually monitoring of exchange reactions of disulfide bonds and discovering the correlation between chemical reactions and mechanical behavior. The paper focuses on the smart material design strategy and mechanics of materials mechanism analysis. The following questions and comments need to be answered before they can be considered for publication:

1. In the introduction, it is recommended to add current or potential applications of the smart surface following lines 55-56.
2. Adding specific applications can improve the work's importance. The pattern is sensitive to heating, and the change process is irreversible. Some cryopreserved items, such as tea, and reagents, may use this anti-counterfeiting icon to indicate good storage and transportation conditions.
3. Robust supramolecular interactions between pyridine and carboxyl underpin the wrinkle structures. Hydrogen bonding is also a clear, dynamic interaction. Whether hydrogen bonding interactions change during the evolution of the wrinkle structure needs to be characterized. Therefore, the bands corresponding to pyridine ($\sim 1600\text{ cm}^{-1}$) and amino-hydrogen ($\sim 3200\text{ cm}^{-1}$) in the SI Fig 9 IR spectrum need to be zoomed in to view. It is also necessary to characterize the changes in hydrogen bonding interactions during the heating process.
4. This work's primary focus is on stress relaxation, which is disputed by the fact that stress relaxation and creep of supramolecular networks should be present simultaneously in the dynamic changes of wrinkles induced by UV light. Dynamic network rearrangement and creep driven by residual stress may erase the wrinkle structure. Therefore, it is recommended to supplement the creep test of the film under UV light.
5. This work analyzes the mechanism of wrinkle evolution from the difference in stress distribution in different directions. Is it reasonable to consider wrinkles parallel to the border in the unexposed area stretching (extending) into the exposed area driven by residual stress and then rearranged? After the heating-cooling cycle, wrinkles parallel to the boundary were then formed.

Reviewer #4 (Remarks to the Author):

The material performance of reconfigurable supramolecular polymer is highly related to molecule-based behavior. In this work, Yan et. al. provided an interesting platform for observing the photo-driven stress relaxation induced by imperceptible perturbations in the intermolecular exchange reaction through wrinkles. The macroscopic sensitive wrinkles as a molecular action detector revealed the chemical-dependent mechanical stress relaxation owing to photo-driven intermolecular exchange, which was quantitatively evaluated by the authors. Just similar to the 'butterfly effect', the trans-scale theoretical models well explained the evolution of ordered microstructures and abrupt transition of wrinkling orientation under imperceptible perturbations. Furthermore, the molecular editing also enables various tunable wrinkle patterns with reprogrammable structures. The resulted reversible pattern of wrinkles is of interest in information encryption and anticounterfeiting. In view of the demonstrated results possessing special phenomenon and insightful mechanism, this work is recommended for publication in Nature Communications. Some minor comments are listed as follow:

- (1) The quantified equations of chemistry-based mechanical stress relaxation connect the intermolecular behavior on a molecular scale and macroscale properties of polymeric networks. Here, I wonder if the isotropic exchange reaction (chemistry) could be affected by the anisotropic boundary (mechanics)?
- (2) As a single exposure region (Fig. 4b), how the different orientations of wrinkle morphology formed at the regions R1 and R1' is influenced by the boundary effect?
- (3) In Fig 4e and Fig. 5b, the color bars seem to be absent.

Response to the Reviewers:

Reviewer #1 (Remarks to Author): In the article by Jiang, Zhang and colleagues under consideration at Nature Communications, the authors report a method to observe dynamic chemistry at interfaces using light. Central to this idea is the interaction of disulfide molecule DTNB that integrates with a PPy-Ba-St polymer film through hydrogel bonding-UV irradiation triggers wrinkles in the film accompanied by disulfide exchange reactions. The authors reflect on how molecular changes have a “butterfly effect” leading to macroscopic phenomena. Intriguingly, the reactions at interfaces are guided by stress relaxation and the gradient properties of the network is shown to propagate after heating/cooling with scope for using this approach to encode information within supramolecular films. This is an exciting work that is carefully done with nice theory, computation and experiment supporting the proposed mechanisms. I believe it warrants publication after considering some minor points as follows:

Response: Thanks a lot for your very positive comments and recommendation. Just as you mentioned, we presented an innovative strategy for visualizing the imperceptible molecular behavior via reprogrammable wrinkle patterns, which opened a new avenue for the trans-scale mechanical mechanism from the intermolecular behavior on a molecular scale to macroscale properties of polymeric networks. And we have addressed your insightful comments as follow:

Comment 1: *The presumed hydrogen bonding between the carboxylate of the DTNB and N of the pyridine are proposed to be key drivers in wrinkle generation. However, there is little evidence that H-bonding is a central mechanism. Hydrogen bonding will also be influenced by temperature changes –could changes in H-bonds after illumination influence molecule mobility and yield the wrinkling orientation behavior? The incorporation of DTNB into the film needs to be demonstrated through supporting experiments (e.g. FTIR/Raman stretching modes with H-bonds, etc.) and discussed in more detail to guide the reader in the molecular characteristics. For*

instance, what is the concentration/density of DTNB in the polymer and how does this effect the observed behavior?

Response to 1: Thank you very much for your professional and constructive comments. We agree with you very much that the hydrogen bonding between pyridine and carboxyl groups is essential for the generation of wrinkles. Actually, as shown in our main text (Page 7, line 179), we have prepared two specimens with different top films (PPy-Ba-St@Phenyl disulfide and PPy-Ba-St@DTNB) to investigate the role of hydrogen bonding in the actuation of the wrinkle patterns. Owing to the lack of crosslinking by hydrogen bonds between pyridine and carboxyl groups, no pattern has been produced on the surface with PPy-Ba-St@Phenyl disulfide as shown in Fig R1 (supplementary Fig. 5), which compared with PPy-Ba-St@DTNB. In addition, we also applied the breakup of hydrogen bonding into wrinkles' erasure by acid gas (HCl), which confirmed the role of hydrogen bonding as well. And the HCl could be released through a thermal treatment so that the wrinkled patterns can be recovered (Fig. R4).

Fig R1. LSCM images and specimens of the top films: (a) PPy-Ba-St@Phenyl disulfide and (b) PPy-Ba-St@DTNB. Scale bar: 100 μm

Upon UV light irradiation (365 nm, 15 mW/cm²), the surface temperature of the PDMS elastomer was monitored by an infrared camera within irradiation for 6 min. The result shows that the temperature changes by a few degrees (Fig. R2). Thus, the hydrogen bonding will not be influenced without temperature changes, and it would

not impact the molecule mobility and yield the wrinkling orientation behavior after illumination influence.

Fig. R2. Temperature variation of PDMS elastomer during 365 nm UV light irradiation with intensity of 15 mW/cm².

Additionally, the effect of DTNB's incorporation into the film was confirmed by the wrinkle patterns in contrast to the film containing PPy-Ba-St (Fig. R3b). The evolution of erasure and generation of the pH-sensitive wrinkle patterns was contributed to the hydrogen bond between pyridine and carboxyl groups (Fig. R4), demonstrating the incorporation of DTNB into the film. As the reviewer's recommended, we further characterized the incorporation of DTNB by FTIR spectrum (Fig. R5), the peak assigned to the COOH stretching vibration (3450 cm⁻¹) shifted to lower wavenumbers (*Macromolecules* 2015, 48, 2022–2029), which is the typical stretching vibration peak of a hydrogen bond.

Fig. R3. LSCM images and specimens of the top films: (a) PPy-Ba-St and (b) PPy-Ba-St@DTNB.

Fig. R4. LCSEM images of wrinkles when the wrinkled samples were exposed to 56.4 ppm HCl vapor for 25 s, and subsequently reheated to release the HCl.

Fig. R5. FT-IR spectrum of DTNB, PPy-Ba-St, and PPy-Ba-St@DTNB.

Considering the concentration/density of DTNB in the polymer, we added detailed information about the formulation of the polymer film in the experimental section accordingly Page 4, line 95-97 of the revised manuscript and page 4, line 68-83 of the revised supporting information. In this work, we have composited PPy-Ba-St and DTNB with different molar ratios for pattern preparation and observation of perturbation behavior. As shown in Fig. R6, too little DTNB will directly lead to the inability to form a wrinkled pattern on the surface (PPy-Ba-St: DTNB=10:0/10:0.5). As the concentration of DTNB increases, the phase separation phenomenon can be observed with inhomogeneous distribution of wrinkled pattern on the surface induced by the excessive cross-linking intensity between pyridine and carboxyl groups. Thus, we employ the molar ratio of 10:1 (PPy-Ba-St: DTNB=10:1) to carry out subsequent experimental studies.

Fig. R6. The evolution of wrinkled morphology with different concentrations of DTNB in the polymer.

Comment 2: *The demonstrations in the latter half of the paper are compelling. One addition that would broaden the scope of the work for the journals broad audience would be some discussion of how these concepts and experiments might be translated to other materials systems. Is this behavior unique to this system or is there some general principles in macromolecular design that could translate to other systems? Some discussion of how other dynamic bonds, polymer systems with different viscoelastic properties, etc., might be amenable to this approach would be beneficial. If this phenomenon is highly specific to this system, this should be stated with some examples of how future studies will yield new knowledge, new devices, etc.*

Response to 2: Thank you very much for the reviewer's insightful comments. Just as you commented, we have presented a bran-new dynamic model for linking the time-varying molecular scale chemical reaction to macroscopic mechanical properties of the wrinkled surface in this work. Actually, these general principles could translate to other systems with appropriate molecular design. The polymer in this work can be replaced with other polymer systems with similar mechanical properties, not even the same constitutional unit. Besides, the disulfide-containing DTNB also can be replaced with any molecular with dynamic bonds on conditions that can be composited to a supramolecular system with the polymer, such as diselenide-containing molecules. In fact, we are exploring this issue and focusing on the molecular action detector through the changing macroscopic surface patterns for imperceptible chemical reactions. The corresponding discussions have been added in the revised manuscript (Page 5, line 121).

Comment 3: *On page 8, line 184 experiments are described using control materials. It is not clear in the main manuscript what the (i) and (ii) materials are without referring to SI. These should be described better in the main text with specific mention to the SI.*

Response to 3: Thank you for your good advices. The sample i only contains pyridine-containing copolymer PPy-Ba-St, sample ii is comprised by PPy-Ba-St copolymer and disulfide-containing molecule (Phenyl disulfide). In addition, the sample iii (PPy-Ba-St@H2MDB) and sample iv (PPy-Ba-St@DTNB) are a supramolecular polymer network through the hydrogen bonding between pyridine and carboxyl groups. The manuscript has been carefully revised according to the suggestions to make these descriptions clear (Page 8, line 181-186 of the revised manuscript).

Comment 4: *The light diffraction patterns shown as insets of Figure 3 are not well explained. More text is needed to explain how this result supports the orientation transition.*

Response to 4: Thanks for your constructive advice. Actually, the diffraction patterned gratings are derived from the light scattering effect of ordered patterns on the surface. The initially disordered topological structures presented a light diffraction pattern with concentric rings (Fig R7), which were viewed by a laser beam (650 nm). After a selective erasure with a striped photomask upon 365 nm UV light, the distinguishing discrete diffraction points were obtained. When the laser passed through the samples without topological structures after heating/cooling treatment, one laser point can be observed on the receiving plane first, and the rectangular diffraction patterns were observed subsequently, which verified the generation of orthogonal wrinkle patterns. The corresponding explanation has been added following line 228-237 (page 9) in the revised manuscript as reviewer suggested.

Fig R7. Schematic demonstration of the diffraction patterned grating based on microstructures.

Reviewer #2 (Remarks to the Author)

To provide a more clearly point-by-point response, we split the comments:

1. *This paper is clearly incremental research compared with papers published from the same group, such as: “Acc. Chem. Res. 2019, 52, 4, 1025-1035”, “Science Advances 2018, 4 (4), eaar5762”, “Adv. Mater. 2020, 32, 1906712”, and “Nat. Commun. 2020, 11 (1), 1811”.* **5.** *Generally, this paper shows a bit of improvement of the “wrinkle pattern” module. I think it won’t attract common interest. Ant the novelty of this paper doesn’t reach the requirement of “Nature Communication”. Thus, I suggest to reject this paper.*

Response to 1 and 5: We thank the reviewer a lot for your efforts in this work, they really inspired us to further improve the quality of the presented manuscript. To be honest, basically, the main points of this proposed work is absolutely different from the previous reports. In recent years, our group devoted ourselves to exploring wrinkled surfaces and have developed a series of smart patterned surfaces, such as NIR-driven wrinkles tuned by applied strain, diffusion-induced 3D patterns, and fluorescent wrinkle dual-pattern. These works relating to physically strain-regulated or chemically modulus-regulated wrinkling patterns were summarized in a Review Article (Ref. 41, *Acc. Chem. Res.* 2019, 52, 1025-1035), laying a foundation for further study of wrinkles. Even if compared to the interesting advances reported in these works, this manuscript as well demonstrated essentially creative, fascinating results among bran-new concepts, attractive phenomena, insightful theory, and potential impacts (as seen in **Table R1**), which has been approved by other reviewers. We not only presented a reliable strategy for observing chemical-induced stress relaxation via wrinkles but also developed a chemical-mechanical trans-scale theoretical model. Besides, the very rare phenomenon that imperceptible disulfide exchange reaction widely used in constructing smart materials can induce significantly changed properties of materials (similar to ‘butterfly effect’) and was clearly demonstrated for the first time.

Table R1. Highlighted advances in this work compared to previous strategies

	Previous strategies	This work
Main concepts	To fabricate dynamic wrinkling surfaces via physical (R2) or chemical stimuli (R3, R4)	To exploit an effective approach for visually observing imperceptible chemical changes and even following chain reactions by microscale dynamic wrinkle.
Feature phenomenon	Dynamic wrinkles with fluorescent, hierarchical 3D (R3), or adjustable microstructures (R2).	Reasonable evolution process of ordered microstructures from random wrinkles and rarely mutational perpendicular orientation of wrinkling topology.
Insightful Mechanism	Focused on mechanical mechanism or constraint -boundary wrinkle model (R1-4)	Quantified chemical-dependent mechanical stress and unique trans-scale theoretical models from chemical to mechanical.
Potential Impacts	 • Fabrication and regulation of wrinkles for various applications. (both) • Quantized presentation of perturbations. (this work) • A constructive bond between chemistry and mechanics, from molecule scale to macroscopic properties. (this work) 	

[R1] *Acc. Chem. Res.* 2019, 52, 4, 1025-1035 (Ref 41 in manuscript)

[R2] *Science Advances* 2018, 4 (4), eaar5762

[R3] *Adv. Mater.* 2020, 32, 1906712 (Ref 24 in manuscript)

[R4] *Nat. Commun.* 2020, 11 (1), 1811 (Ref 33 in manuscript)

(1) The listed reference R1 (*Acc. Chem. Res.* 2019, 52, 1025-1035 Ref 41 in this manuscript) is a **review article**, which illustrates the main strategies and highlights the important recent progress on smart patterned surfaces with dynamic wrinkles, including their design, preparation, and potential applications. The dynamic wrinkling patterns could be realized by physically (applied stress) or chemically (thermal Diels-Alder reaction, photoreaction, or supramolecular chemistry) driven strategies. The unique feature of our work might be directly found in contrast to mentioned multidimensional strategies. Herein, neither the intermolecular perturbation-induced variable microstructures and trans-scale mechanism of interesting phenomena nor the

cutting-edge concept of visualized chemical-triggered ‘butterfly effect’ has been mentioned in this systematic overview whatever from summary to outlook, which can exactly prove the presented work is innovative.

(2) In the second listed work R2 (*Science Advances* 2018, 4, eaar5762), Li *et al.* reported a physically approach for fabricating near-infrared (NIR) light-responsive dynamic wrinkles by utilizing carbon nanotube (CNT)-containing PDMS elastic substrate, the wrinkling surfaces were physically tuned by regulating the applied strain of the bilayer systems in a general manner. Apparently, there are fundamental differences between the two approaches. In this work, we creatively utilized the sensitive wrinkle as a molecular action detector for revealing unobservable chemical-dependent mechanical stress relaxation, and reflected on how molecular changes have ‘butterfly effect’ leading to the macroscopic phenomenon. And that would explain why we did not cite this second listed work.

(3) The third listed literature R3 (*Adv. Mater.* 2020, 32, 1906712, Ref 24 in this manuscript), Li *et al.* developed a strategy for fabricating hierarchical 3D patterns with reversible wrinkles on the top layer using a Diels-Alder reaction. Selective photodimerization of the maleimide resulted in directed diffusion. On the contrast, our strategy in this manuscript can record and demonstrate the imperceptible chemical reactions. Even if excluding the aforementioned features and advances of the proposed strategy, it is certain that not only the variable orientation of wrinkles induced by imperceptible intermolecular perturbations has nothing to do with diffusion-caused patterns, but also the presented trans-scale chemical-mechanical model for the first time is totally different from the theoretical boundary effect modeling, such is one of the highlights of our work.

(4) And in the last listed reference R4 (*Nat. Commun.* 2020, 11, 1811, Ref 33 in this manuscript), Ma *et al.* focused on demonstrating a dynamic dual-pattern with tunable wrinkles and fluorescence for fabricating responsive patterns. It is an interesting work, but did not include any theoretical wrinkling modes. Basically, the real correlation between the two works is weak. The evolution process of wrinkling surfaces was

significantly different. In comparison, this manuscript here quantified the reaction-induced stress and bridged the gap between anisotropic chemical exchange and inhomogeneous stress distribution, which may have more profound impacts in the fields of chemistry and mechanical mechanics.

In this manuscript, we not only pursued tuning wrinkles based on photo-regulated molecule editing and fabricated interesting reprogrammable patterns. Maybe there are some seemingly similar results possessing erasable ordered or perpendicular wrinkling microstructures, more importantly, to the best of our knowledge, this is the first presentation for *in-situ* observing dynamic stress relaxation induced by chemical reaction through sensitive wrinkles, and establishing trans-scale theoretical models, opening an avenue for understanding and meeting multidisciplinary requirements.

2. *Especially, the mechanical model has been shown in “Adv. Mater. 2020, 32, 1906712” (Figure 3).*

Response to 2: This work is fundamentally different from previous work in terms of the mechanical models (Table R2). In this paper, we have proposed a bran-new trans-scale mechanical model (Section S3 in supporting information), which can reveal the relationship between molecular motion (ultraviolet-induced dynamic intermolecular behavior) and macroscopic mechanics (macroscopic mechanical properties of wrinkled DTNB/PDMS two-layer systems). As indicated in Eq. (1) and supplementary section 3.1, photo-driven disulfide exchange (molecular scale) can induce the time-varying stress relaxation in the top film and furthermore the dynamic evolution of the wrinkled morphology (microscale). However, the finite constraint-boundary wrinkle model in Ref 24 (*Adv. Mater.* 2020, 32, 1906712) is a continuum mechanical model, not a trans-scale model, which is unable to reveal the trans-scale mechanical mechanism between molecular behavior and wrinkling morphology. In addition, the finite constraint-boundary wrinkle model is used for explaining the 3D hierarchical wrinkling patterns in Ref 24. In this manuscript, a new trans-scale mechanical model is established to reveal 2D wrinkling pattern evolution

mechanism caused by photodynamic chemical reaction-induced intermolecular ‘butterfly effect’. What's more, the presented trans-scale chemical-mechanical model is totally different from the theoretical boundary effect model, such is one of the highlights of our work. We believe that the trans-scale mechanical model in our work is of great scientific significance for deepening the understanding of dynamic chemistry-induced smart surface science in the future.

Table R2. The compared theoretical mechanism model.

Model	Scale	Scope	Constitutive relation
Li. [Adv. Mater. 2020, 32, 1906712, Ref 24]	Macroscopic continuum mechanical model	3D hierarchical wrinkling patterns 	$\sigma_{xx} = \sigma_0 + \bar{E}_{fc} \frac{du_x}{dx}$ $\sigma_{yy} = \sigma_0 + \nu_f \bar{E}_{fc} \frac{du_x}{dx}$ (macroscopic stress and strain in top film)
This work	Trans-scale mechanical model 	2D wrinkling patterns 	$\Delta\sigma(t) = kT_a Nv \left(\frac{r_0}{NB}\right)^2 [\theta^2(t) - 1/\theta^2(t)]$ (macroscopic stress in the top film and molecular scale photo-driven disulfide exchange)

3. *The authors didn't review these papers properly and comment the difference between this paper, some of which were not even cited.*

Response to 3: Thanks for the reviewer's comments. Tuning the chemical reaction-induced modulus (E_f) and thermal-induced stress (ε) can promote the development of regulated wrinkling surfaces, according to the following equations (Eq. R1 and Eq. R2). In this work, we have demonstrated a totally different manner. Owing to the uninvolved regulation of photo-thermal applied stress, it might be reasonable for us to did not cite the NIR-driven wrinkles (*Sci. Adv.* 2018, 4, eaar5762). Considering the reviewer's comments, we have carefully checked and revised the

review to further highlight advances achieved in our work, and feel that the references have been properly cited and the manuscript has been further strengthened as a result (Page 3, line 59).

$$\ln(A^2 + h_f^2) = \frac{2}{3} \ln \overline{E}_f + \left(2 \ln h_f + \ln 4\varepsilon - \frac{2}{3} \ln 3\overline{E}_s \right) \quad (1)$$

$$\varepsilon = (\partial_s - \partial_f) \Delta T \quad (2)$$

4. The authors claim that “wrinkle patterns could be repeatedly written/erased”. But this phenomenon has been reported in several papers.

Response to 4 and 5: Thanks for your comment. Releasing the stress is an interesting key point in many fields. It is worth noting that we are not just focused on fabricating wrinkling patterns in this work. And the several reported erasable wrinkling patterns, as the reviewer said, were obtained here on the basis of visualization and quantification of chemical reaction-induced stress relaxation. The repeatedly written/erased wrinkling patterns also indicated the reliability and repeatability of the proposed strategy. By the way, the evolution process of wrinkling microstructures from random labyrinthic to the ordered arrangement was clearly described and revealed. Furthermore, compared to the previous reported work, aside from realizing rarely mutational orthogonal morphology of patterned surfaces via regulated minute molecule editing and triggered ‘butterfly effect’, more intriguingly, we here for the first time utilized sensitive wrinkling structures for observing chemical exchange and revealed the trans-scale theoretical evolution from chemistry to mechanics. Before this work was presented, we might be puzzled by the following questions: why the imperceptible perturbations can cause ‘butterfly effect’; how to reflect the unimpressive chemistry or other changes in a visual manner; how can chemical reaction and mechanics influence each other. Herein, we provided a facile yet reliable example for understanding/addressing these interesting questions.

Reviewer #3 (Remarks to the Author): *This article constructs a supramolecular cross-linked polymer network, which can undergo rearrangement through UV-induced disulfide bond exchange. A dynamic wrinkled pattern based on the material allows the visually monitoring of exchange reactions of disulfide bonds and discovering the correlation between chemical reactions and mechanical behavior. The paper focuses on the smart material design strategy and mechanics of materials mechanism analysis. The following questions and comments need to be answered before they can be considered for publication:*

Response: We thank you for the very positive comments. As you comment, we have presented a bran-new dynamic model for discovering the correlation between chemical reactions and mechanical behavior. And the clear relationship between chemistry, mechanics, and microstructures was revealed here. The manuscript has been carefully revised according to the suggestions.

Comment 1: *In the introduction, it is recommended to add current or potential applications of the smart surface following lines 55-56.*

Response to 1: Thanks for your good advice. As the reviewer mentioned, the surface with dynamic patterns can promote the development of smart surface, which has been widely found in artificial materials with various functional applications, such as smart displays, structural colors, electronic devices, microfluidic channels, and interface engineering (Page 2, line 45-49). Inspired by the reviewer's advices, we have added more related references of the smart surface in the introduction to arouse a broad interest from readership in the revised manuscript (Page 2, line 45, line 58-60).

Comment 2: *Adding specific applications can improve the work's importance. The pattern is sensitive to heating, and the change process is irreversible. Some cryopreserved items, such as tea, and reagents, may use this anti-counterfeiting icon to indicate good storage and transportation conditions.*

Response to 2: Thanks for your good suggestions. We agree with you very much that the specific applications of our system for indicating goods storage and transportation conditions can be achieved by the sensitive topological structures. As suggested by the reviewer, for example, we could prepare some interesting anti-counterfeiting labels for the medicines that are stable at low temperatures but becomes unstable at high temperatures (Fig. R8). The thermal responsiveness of wrinkles can be used to monitor whether the medicine experienced a high temperature. In general, the morphology of the wrinkles will return to the initial state due to residual stress, so they cannot realize such interesting application. Here, when the medicine was heated to $>70\text{ }^{\circ}\text{C}$, the orientation of the wrinkles obviously changed, indicating that the medicine had the risk of losing its activity at high temperatures. We have prospected these potential applications in the revised manuscript following line 410 (page 16) and feel that the manuscript has been strengthened as a result according to your insightful comments.

Fig. R8. Transformable wrinkles serving as thermal-responsive anti-counterfeiting labels for medicine.

Comment 3: *Robust supramolecular interactions between pyridine and carboxyl underpin the wrinkle structures. Hydrogen bonding is also a clear, dynamic interaction. Whether hydrogen bonding interactions change during the evolution of the wrinkle structure needs to be characterized. Therefore, the bands corresponding to pyridine ($\sim 1600\text{ cm}^{-1}$) and amino-hydrogen ($\sim 3200\text{ cm}^{-1}$) in the SI Fig 9 IR spectrum need to be zoomed in to view. It is also necessary to characterize the changes in hydrogen bonding interactions during the heating process.*

Response to 2: Thank you very much for the reviewer's insightful comments. We have characterized the changes of hydrogen bonding interactions between pyridine and carboxyl groups during the heating process by temperature-dependent FT-IR spectra. As shown in Fig. R9, the corresponding amplified peak assigned to the COOH stretching vibration (3450 cm^{-1}) weakened and shifted to higher wavenumbers with the temperature increased, which is the typical temperature-sensitive behavior of a hydrogen bond. Interestingly, the corresponding peak can return to their original state after cooling down to the room temperature, confirming the reversibility of hydrogen bonds in the top layer (*Macromolecules* 2015, 48, 2022–2029).

Fig. R9. Temperature dependent FT-IR spectra of PPy-Ba-St@DTNB. The experimental temperature was 25, 45, 65, 85, 100 °C, respectively.

Comment 4: *This work's primary focus is on stress relaxation, which is disputed by the fact that stress relaxation and creep of supramolecular networks should be present simultaneously in the dynamic changes of wrinkles induced by UV light. Dynamic network rearrangement and creep driven by residual stress may erase the wrinkle structure. Therefore, it is recommended to supplement the creep test of the film under UV light.*

Response to 4: Thank you very much for the reviewer's valuable comments. As suggested by the reviewer, we have tested the creep of supramolecular networks by

dynamic thermomechanical analysis (DMA). As the results in the DMA curve (Fig. R10a), the creep in the PPy-Ba-St@DTNB specimen shows no remarkable change after sequential exposure to UV light. However, it presented a significant stress relaxation after sequential exposure to UV light in Fig. R10b. Therefore, the erasure of wrinkles is mainly attributed to the stress relaxation by disulfide exchange reaction, which ruled out the erasure of the wrinkle structure by creep.

Fig. R10. The viscoelastic behavior of supramolecular films. (a) creep curves and (b) stress relaxation curves of PPy-Ba-St@DTNB with (yellow line) and without (blue line) UV irradiation.

Comment 5: *This work analyzes the mechanism of wrinkle evolution from the difference in stress distribution in different directions. Is it reasonable to consider wrinkles parallel to the border in the unexposed area stretching (extending) into the exposed area driven by residual stress and then rearranged? After the heating-cooling cycle, wrinkles parallel to the boundary were then formed.*

Response to 5: Thank you very much for the reviewer's insightful comments. It is reasonable to consider wrinkles parallel to the border in the unexposed area stretching (extending) into the exposed area driven by residual stress and then rearranged. In the course of the experiment, we have observed and recorded these interesting phenomena, the dynamic evolution process is shown in Fig R11. In addition, these experimental results have been verified by the theoretical analysis. Owing to the boundary constraint effect (Ref. 28, *Angew. Chem. Int. Ed.* 2016, 55, 3931-3935), the wrinkled pattern was going to rearrange to adapt itself to the light-defined boundary.

Before the heating-cooling cycle, the 1D stress relaxation occurred at the boundaries of the unexposed area (Fig. R12), indicating that the wrinkles will be rearranged and finally perpendicular to the border. At the boundaries, the stress at the y direction is slightly larger than the x direction ($\sigma_y > \sigma_x$), according to Eq. (2). Because of the homogeneous heating-cooling cycle, the stress at the y direction is still larger than the x direction, indicating that the 1D ordered wrinkles along the y direction will be triggered before the wrinkles along the x direction in the exposed area (Fig. R11). For the unexposed region, the elastic modulus of the surface film is slightly smaller than the hard constraint boundary of the exposed region, $E_{ue} = E_0 < E_e = E$, indicating that the wrinkles will be triggered to parallel to the boundary. These results again confirmed the chemical reaction, stress field, and topological microstructures could influence each other, further demonstrating the intriguing feature of our work.

Fig. R11. The dynamic evolution process of wrinkles with selective erasure and heating/cooling treatment.

Fig. R12. The modeling of 1D stress relaxation of the unexposed region is triggered by the disulfide bond exchange reaction of the exposed region.

Reviewer #4 (Remarks to the Author): The material performance of reconfigurable supramolecular polymer is highly related to molecule-based behavior. In this work, Yan et. al. provided an interesting platform for observing the photo-driven stress relaxation induced by imperceptible perturbations in the intermolecular exchange reaction through wrinkles. The macroscopic sensitive wrinkles as a molecular action detector revealed the chemical-dependent mechanical stress relaxation owing to photo-driven intermolecular exchange, which was quantitatively evaluated by the authors. Just similar to the ‘butterfly effect’, the trans-scale theoretical models well explained the evolution of ordered microstructures and abrupt transition of wrinkling orientation under imperceptible perturbations. Furthermore, the molecular editing also enables various tunable wrinkle patterns with reprogrammable structures. The resulted reversible pattern of wrinkles is of interest in information encryption and anticounterfeiting. In view of the demonstrated results possessing special phenomenon and insightful mechanism, this work is recommended for publication in *Nature Communications*. Some minor comments are listed as follow:

Response: We thank you for the very positive comments. As you mentioned, we have established the trans-scale theoretical models for connecting the chemical reaction and mechanical behavior of supramolecular systems. We agree with you that our work has demonstrated novel concepts, special phenomenon, and insightful mechanism, and anticipate that this work can arouse a broad interest from various fields. The manuscript has been carefully revised according to the suggestions.

Comment 1: *The quantified equations of chemistry-based mechanical stress relaxation connect the intermolecular behavior on a molecular scale and macroscale properties of polymeric networks. Here, I wonder if the isotropic exchange reaction (chemistry) could be affected by the anisotropic boundary (mechanics)?*

Response to 1: Thank you very much for the reviewer's insightful comments. The isotropic exchange reaction (chemistry) could be affected by the anisotropic boundary (mechanics). Such as the 1D stress relaxation (mechanics) occurred at the boundaries of the unexposed area, resulting in the labyrinthic wrinkles rearranged to 1D ordered

wrinkles along the y direction by the chain reactions in supramolecular systems. Such results indicated one of the unique highlights of our work.

Comment 2: *As a single exposure region (Fig. 4b), how the different orientations of wrinkle morphology formed at the regions R_1 and R_1' is influenced by the boundary effect?*

Response to 2: Thank you very much for the reviewer's valuable comments. Region R_1 is the first-exposed region, which is constrained by the soft region (unexposed region) along the x direction and hard region (double-exposed region) along the y direction, therefore generating the ordered wrinkles along the y direction. On the contrary, the orientations of the wrinkle in region R_1' is along the x direction. It is because the constrained boundaries of this second-exposed region have the opposite direction compared to the region R_1 .

Comment 3: *In Fig 4e and Fig. 5b, the color bars seem to be absent.*

Response to 3: Thank you for kindly reminding us in this paper. We have marked the color bars (Fig. 4e and Fig. 5b) and carefully checked the revised manuscript.

Reviewer comments, second round –

Reviewer #1 (Remarks to the Author):

The authors have adequately responded to the concerns raised in my initial review with some change to the text and inclusion of new data. Once the authors have satisfactorily addressed the concerns of all reviewers, I believe this manuscript will make a nice addition to the journal.

Reviewer #4 (Remarks to the Author):

In the revised version, the main concerns have been well addressed. I recommend the acceptance of the manuscript for the publication by Nat. Commun.

Reviewer #1 (Remarks to the Author): The authors have adequately responded to the concerns raised in my initial review with some change to the text and inclusion of new data. Once the authors have satisfactorily addressed the concerns of all reviewers, I believe this manuscript will make a nice addition to the journal.

Response: Thank you again for your positive comments.

.

Reviewer #4 (Remarks to the Author): In the revised version, the main concerns have been well addressed. I recommend the acceptance of the manuscript for the publication by Nat. Commun.

Response: Thank you again for your positive comments.